# Endoscopic Images by a Single-Shot Multibox Detector for the Identification of Early Cancerous Lesions in the Esophagus: A Pilot Study

**DOI:** 10.3390/cancers13020321

**Published:** 2021-01-17

**Authors:** Yao-Kuang Wang, Hao-Yi Syu, Yi-Hsun Chen, Chen-Shuan Chung, Yu Sheng Tseng, Shinn-Ying Ho, Chien-Wei Huang, I-Chen Wu, Hsiang-Chen Wang

**Affiliations:** 1Division of Gastroenterology, Department of Internal Medicine, Kaohsiung Medical University Hospital, Kaohsiung Medical University, No.100, Tzyou 1st Rd., Sanmin Dist., Kaohsiung City 80756, Taiwan; fedwang@gmail.com (Y.-K.W.); jayshung1985@gmail.com (Y.-H.C.); 2Graduate Institute of Clinical Medicine, College of Medicine, Kaohsiung Medical University, No.100, Tzyou 1st Rd., Sanmin Dist., Kaohsiung City 80756, Taiwan; 3Department of Medicine, Faculty of Medicine, College of Medicine, Kaohsiung Medical University, No.100, Tzyou 1st Rd., Sanmin Dist., Kaohsiung City 80756, Taiwan; 4Department of Mechanical Engineering and Center for Innovative Research on Aging Society (CIRAS), National Chung Cheng University, 168, University Rd., Min Hsiung, Chia Yi 62102, Taiwan; shihoi543@gmail.com (H.-Y.S.); lonesome310160@hotmail.com (Y.S.T.); 5Department of Internal Medicine, Far Eastern Memorial Hospital, No.21, Sec. 2, Nanya S. Rd., Banciao Dist., New Taipei City 220, Taiwan; chungchenshuan_3@yahoo.com.tw; 6Institute of Bioinformatics and Systems Biology, National Chiao Tung University, Hsinchu City 30010, Taiwan; syho@mail.nctu.edu.tw; 7Department of Gastroenterology, Kaohsiung Armed Forces General Hospital, 2, Zhongzheng 1st Rd., Lingya District, Kaohsiung City 80284, Taiwan; 8Department of Nursing, Tajen University, 20, Weixin Rd., Yanpu Township, Pingtung County 90741, Taiwan

**Keywords:** esophageal cancer, single-shot multibox detector, artificial intelligence, convolutional neural network

## Abstract

**Simple Summary:**

Detection of early esophageal cancer is important to improve patient’s survival, but accurate diagnosis of superficial esophageal neoplasms is difficult even for experienced endoscopists. Computer-aided diagnostic system is believed to be an important method to provide accurate and rapid assistance for endoscopists in diagnosing esophageal neoplasms. We developed a single-shot multibox detector using a convolutional neural network for diagnosing esophageal cancer by using endoscopic images and the aim of our study was to assess the ability of our system. Our system showed good diagnostic performance in detecting as well as differentiating esophageal neoplasms and the accuracy can achieve 90%. Differentiating different histological grades of esophageal neoplasm is usually conducted by magnified endoscopy and we confirm that artificial intelligence system has great potential for helping endoscopists in accurately diagnosing superficial esophageal neoplasms without the necessity of magnified endoscopy and experienced endoscopists.

**Abstract:**

Diagnosis of early esophageal neoplasia, including dysplasia and superficial cancer, is a great challenge for endoscopists. Recently, the application of artificial intelligence (AI) using deep learning in the endoscopic field has made significant advancements in diagnosing gastrointestinal cancers. In the present study, we constructed a single-shot multibox detector using a convolutional neural network for diagnosing different histological grades of esophageal neoplasms and evaluated the diagnostic accuracy of this computer-aided system. A total of 936 endoscopic images were used as training images, and these images included 498 white-light imaging (WLI) and 438 narrow-band imaging (NBI) images. The esophageal neoplasms were divided into three classifications: squamous low-grade dysplasia, squamous high-grade dysplasia, and squamous cell carcinoma, based on pathological diagnosis. This AI system analyzed 264 test images in 10 s, and the sensitivity, specificity, and diagnostic accuracy of this system in detecting esophageal neoplasms were 96.2%, 70.4%, and 90.9%, respectively. The accuracy of this AI system in differentiating the histological grade of esophageal neoplasms was 92%. Our system showed better accuracy in diagnosing NBI (95%) than WLI (89%) images. Our results showed the great potential of AI systems in identifying esophageal neoplasms as well as differentiating histological grades.

## 1. Introduction

Esophageal cancer is a highly aggressive cancer with a poor prognosis, and around 508,000 esophageal cancer-related deaths were recorded globally in 2018. It is also the seventh most common cancer and the sixth most common cause of cancer-related death [1]. The prognosis of esophageal cancer is usually good in its early stages with a 5-year survival rate reaching 80%, but extremely poor in its advanced stages with a 5-year survival rate of less than 20% [2]. However, most esophageal cancer is diagnosed at advanced stages because typical symptoms such as dysphagia and odynophagia usually develop during these later stages.

Esophagogastroduodenoscopy (EGD) is the most sensitive examination approach as well as being the gold standard for diagnosis of esophageal cancer and precancerous lesions; nevertheless, the diagnosis of esophageal precancerous lesions and superficial cancer still presents great challenges for endoscopists, as these lesions are easily overlooked in conventional white-light imaging (WLI) and about 40% of lesions might be missed [3]. Even though image-enhanced endoscopy, such as Lugol’s chromoendoscopy and narrow-band imaging (NBI), is recommended in addition to WLI to improve the detection rate of esophageal precancerous lesions as well as superficial cancer, interobserver variation still exists, especially with inexperienced endoscopists [4,5,6].

With the development of computer technology, artificial intelligence (AI) has been widely studied in the endoscopic field in diagnosing gastrointestinal tract diseases, especially cancer [7]. One major role of computer-aided diagnosis is to help endoscopists in differentiating between neoplastic and non-neoplastic lesions, and several studies have proven the potential of AI systems in the diagnosis of early esophageal cancer, including squamous cell carcinoma and adenocarcinoma [8,9,10,11]. Furthermore, the diagnostic accuracy of AI systems has also been compared with experienced endoscopists, and comparable performance has been reported [12,13,14]. However, most studies have evaluated the diagnostic accuracy of AI systems by using two nominal variables (non-cancer and cancer), and a few studies have used ordinal variables to evaluate different histological grades of esophageal neoplasms such as low-grade dysplasia, high-grade dysplasia, and cancer, by AI systems. In our previous study, we found significant spectral differences of endoscopic images between normal, precancerous, and cancerous lesions of the esophagus by a computer-aided system [15]. For further analysis of the ability of AI systems in differentiating the histological grade of esophageal neoplasms, we developed a deep learning system using a single-shot multibox detector (SSD) for image recognition. SSD is a deep convolutional neural network (CNN) consisting of 16 layers or more, and CNN is known as one of the best performance models of AI systems in image recognition [16,17].

In the present study, we aimed to test the ability of an AI-assisted image analysis system in differentiating histological grades of esophageal neoplasms, including low-grade squamous dysplasia, high-grade squamous dysplasia, and squamous cell carcinoma (SCC).

## 2. Results

### 2.1. Diagnostic Performance of Our AI System for Detecting Esophageal Neoplasm

A total of 264 images were used as a test image set, including 112 WLI and 152 NBI images, and SSD required 10 s for analysis. The comprehensive SSD accurately diagnosed 202 images of 210 images of esophageal neoplasms and 38 images of 54 images of a normal esophagus (Table 1). The sensitivity, specificity, positive predictive value (PPV), negative predictive value (NPV), and accuracy of comprehensive SSD for esophageal neoplasms were 96.2%, 70.4%, 92.7%, 82.6%, and 90.9%, respectively (Table 2). After comparing the diagnostic performance between WLI and NBI, SSD showed higher specificity and PPV in diagnosing WLI images as compared with NBI. In contrast, NBI images provided higher sensitivity and NPV. However, the accuracy of SSD was similar in diagnosing NBI and WLI images (*p* = 0.61).

### 2.2. Diagnostic Performance of Our AI System for Differentiating Histological Grade of Esophageal Neoplasm

The detailed results of our SSD in analyzing different histological grades of esophageal neoplasms are shown in Table 3. The diagnostic accuracy of comprehensive SSD was 92%, and the SSD showed higher accuracy in diagnosing NBI images (95%) than WLI images (89%). The kappa values for WLI and NBI were 0.82 (95% confidence interval = 0.66–0.97) and 0.91 (95% confidence interval = 0.77–1.05), respectively. Our SSD showed good sensitivity for esophageal SCC, and the sensitivity of comprehensive SSD, WLI, and NBI for esophageal cancer was 98.9%, 97.5%, and 100%, respectively. We found better sensitivity, PPV, and F1-score of our SSD in analyzing NBI images of different histological grades of esophageal neoplasms than WLI (Table 4). A trend of better diagnostic performance on PPV and F1 score in advanced malignant lesions, rather than low-grade dysplasia, was also demonstrated in our comprehensive analysis model.

## 3. Discussion

Our AI system showed good diagnostic performance for the detection of esophageal neoplasms, and the accuracy, sensitivity, and PPV of our SSD were 90.9%, 96.2%, and 92.7%, respectively. Previous studies using other AI systems in diagnosing esophageal neoplasms showed the accuracy, sensitivity, and PPV of different AI systems were about 56–93%, 89–98%, and 46–86%, respectively [8,11,18,19,20]. Our system, therefore, demonstrated compatible results with these previous studies. The diagnostic performance of our SSD in analyzing WLI and NBI images was tested, but no apparent increase in diagnostic accuracy was observed in analyzing NBI images. A previous study showed no significant difference was found between using WLI and NBI images for AI diagnosis [20]. Although NBI increased the sensitivity by magnifying the features of neoplasm, NBI might also decrease the specificity through over-diagnosis. Similar findings (higher sensitivity and lower specificity) were also observed in a previous prospective study evaluating the efficacy of NBI in the diagnosis of esophageal lesions by endoscopists [21].

In addition to the detection of esophageal neoplasm, our SSD system also demonstrated the capacity to differentiate histological grades of neoplasms into low-grade dysplasia, high-grade dysplasia, and cancer. The accuracy of our SSD system for the diagnosis of histological grades of esophageal neoplasm was 92%, and NBI (95%) showed higher accuracy than WLI (89%). To the best of our knowledge, this study is the pilot study using an AI system to differentiate between histological grades of esophageal neoplasm, and a diagnostic system that not only detects esophageal neoplasm but also identifies the histological grade of esophageal neoplasm was constructed.

Most previous studies used two nominal classifications in testing the diagnostic performance of an AI system for esophageal neoplasm, including cancer and non-cancer or a region of interest and a region of non-interest [8,10,14,20]. Some other studies have used an AI system in evaluating the invasion depth of esophageal cancer, and two nominal classifications for invasion depth were used, including submucosal microinvasion and submucosal deep invasion [13,22]. A few studies have used more than two nominal classifications to test the ability of an AI system in diagnosing esophageal neoplasms. In the present study, we found NBI showed better diagnostic accuracy in differentiating esophageal neoplasms than WLI. Although no previous study could be compared, NBI did show better accuracy than WLI in lesion detection in previous endoscopic screening studies for esophageal neoplasms [23]. Actually, magnified endoscopy with NBI had an important role in evaluating the invasion depth of esophageal neoplasm, and the differentiation between low-grade dysplasia, high-grade dysplasia, and cancer might have been achieved by experienced endoscopists [24]. Our result showed the potential benefit of an AI system in differentiating histological grades of esophageal neoplasm without the limitations of the requirement of magnified endoscopy and, more importantly, the experience of endoscopists.

In this pilot study, we found the severity of histological grade influenced the diagnostic performance of our AI system. A trend of diagnostic performance of comprehensive analysis between cancer and low-grade dysplasia was observed; our AI system showed the highest PPV and F1 score in diagnosing cancer, on the other hand, low-grade dysplasia had the lowest PPV and F1 score. The same finding was also observed in the WLI model. Discussing the reasons as to why histological grade influences the diagnostic performance of the AI system is problematic because deep learning did not explain the predictions; however, this phenomenon was not observed in the NBI model. Our explanation is that the classification of esophageal neoplasms is made by pathological diagnosis, and it is supposed that some image features of advanced esophageal neoplasms could be identified by the AI system in the WLI model, while the image differences between early and advanced lesions would be diminished by the NBI model as NBI magnifies some features that could not be demonstrated on WLI.

The specificity rate of our AI system was 70.4%, and previous studies have shown specificity rates of about 68–96% [8,11,13,14,20,22]. After analyzing the causes of a false-positive result, we found possible reasons including fewer training images of a normal esophagus, the quality of the image was not sharp enough, or the shadow of the esophagus affected the SSD diagnosis. A previous study also found the most common cause of false-positives was a shadow, followed by normal structure and benign lesion [20]. Our study used only 936 images as the training set for the construction of our SSD system, and this number was far less than previous studies where the AI system used 8000–10,000 images for training. Although a key feature of the SSD model is the multiscale convolutional bounding-box output using multiple feature maps, this model did not have to learn image boundary features that were too complicated; however, more training images or carefully selected frames might still be needed to improve the diagnostic performance of our system.

There are several limitations in our study. Firstly, our SSD system was constructed by images from a single center using fewer training images as compared to previous studies. In addition, images were still used from regular endoscopes (GIF-Q260; Olympus Medical Systems, Co, Ltd., Tokyo, Japan) and standard endoscopic video systems (EVIS LUCERA CV-260/CLV-260; Olympus Medical Systems, Co, Ltd., Tokyo, Japan), rather than non-high-resolution endoscopy images and system. Whether the diagnostic performance of our SSD system could be improved by higher quality images from a newer model machine or magnified endoscopy is unknown; however, based on the results of the present study, our SSD system shows non-inferior diagnostic performance in detecting esophageal neoplasms and has a great potential in differentiating neoplasms into different histological grades. Furthermore, a newer machine or magnified endoscopy is not necessary for our system, and this advantage might be of great help in institutions without these devices. Secondly, as mentioned before, fewer images of a normal esophagus were used in our training set and data imbalance between different groups might have interfered with the model optimization; moreover, the number of images from each lesion was not consistent, and this might have also influenced the learning effect of our SSD system. Thirdly, suboptimal images with poor quality were excluded in both training and testing sets, which might have caused selection bias; and fourthly, we focused mainly on esophageal squamous dysplasia and squamous cell carcinoma, and images of Barrett’s esophagus and esophageal adenocarcinoma were not collected for analysis. Because of the small sample size, we believe that constructing an AI system merely focused on squamous cell neoplasms would provide better diagnostic accuracy. Further study might be warranted to evaluate our SSD system in diagnosing Barrett’s esophagus as well as esophageal adenocarcinoma.

## 4. Materials and Methods

### 4.1. Study Design and Preparation of Training and Test Image Sets

To construct our SSD (SSD-HS, Hitspectra Intelligent Technology Co., Ltd., Kaohsiung City, Taiwan) system for the diagnosis of esophageal neoplasm, we retrospectively collected EGD images from 46 patients with esophageal neoplasms at Kaohsiung Medical University Hospital, and there were 10 patients with esophageal low-grade dysplasia, 20 patients with high-grade dysplasia, and 16 patients with SCC. The classification of esophageal neoplasm was based on pathological reports. A total of 936 images were collected for training images, including 162 images of a normal esophagus, 165 images of low-grade squamous dysplasia, 282 images of high-grade squamous dysplasia, and 327 images of esophageal cancer (SCC). The number of WLI and NBI images was 498 and 438, respectively. An additional 264 images were also obtained for the test set, and these images included 112 WLI and 152 NBI images. We excluded low-quality images caused by blurring, defocusing, mucus, and poor air blowing. This study was approved by the Institutional Review Board of Kaohsiung Medical University Hospital (KMUH) (KMUHIRB-E(I)-20180338). Written informed consent was waived because of the retrospective, anonymized nature of study design.

### 4.2. Construction of AI System

A convolutional neural network (CNN) architecture, called the single-shot multibox detector (SSD) model, was used in constructing a diagnosis system based on AI [25,26,27,28,29]. It is a fast object detector for multiple categories within one stage, as shown in Figure 1.

The crucial feature of SSD is that it can provide at least one order of magnitude and a default box with different locations, scales, and aspect ratios compared with the existing methods; additionally, the SSD architecture utilizes multiscale convolutional bounding-box output with multiple feature maps [27,30]. The setting of the default box is shown in Figure 2. Overly complicated image boundary features are not needed for model training, and SSD can be efficiently trained to learn the possible boundary box dimensions. The SSD filters were fine-tuned using stochastic gradient descent.

Several rectangular default boxes with different sizes and positions can be obtained through the settings above. However, the predicted results of the model output only needed a few boundary boxes to match the ground truth. Therefore, SSD required a mechanism that could match or eliminate redundant default boxes. The matching principles between the default boxes and the ground truth are defined as follows. First, how the ground truth of the training images matches the default boxes must be determined. The matching degree between the default boxes and the ground truth is determined by calculating the Intersection over Union (IOU) value and used in ensuring that each ground truth corresponds to a unique default box [31]. The bounding box corresponding to the prior box matching the ground truth is responsible for the prediction. The IOU value is between 0 and 1, and a large value indicates a high matching degree between the default box and ground truth. The IOU value of the prediction box and ground truth is ideally 100%. Figure 3 shows the principle of how a default box matches a ground truth. Figure 3a demonstrates that the SSD forms several default boxes to match the ground truth. The two main principles of matching are as follows. The first principle is that for each ground truth, a default box matching with the largest IOU value must be present to guarantee that each ground truth can match a certain default box, as shown in Figure 3d. The default box that matches the ground truth is called a positive sample, which eventually becomes a boundary box. If a default box does not match any ground truth, it can only match the background, which is called the negative sample, as shown in Figure 3b. Ground truths are few in an image, whereas default boxes are numerous. If the first principle is adopted to match the ground truth, many default boxes become negative samples, and the ratio between positive and negative samples becomes unbalanced. Therefore, the second principle is needed: for the rest of the unmatched default boxes, if the IOU value of a ground truth is greater than a specific threshold (generally 0.5), then the default box matches the ground truth, as shown in Figure 3c. The second principle means that a certain ground truth may match several default boxes; by contrast, a default box can only match a ground truth. If multiple ground truths match a certain default box and their IOU values are all greater than the threshold, the default box only matches the ground truth with the largest IOU value. The second principle is performed only after the first principle, and the output results are shown in Figure 3e. After the SSD was used to learn the training image set, 264 independent test images were used in evaluating the performance of the trained model.

When the model detector detects an esophageal neoplasm from the input data of the test image, the disease name (normal, low-grade dysplasia, high-grade dysplasia, or cancer) is assigned, and a rectangular frame is displayed in the endoscopic image to surround the area of the esophageal neoplasm. In addition, if the area is normal, the rectangular frame is not displayed. Figure 4 and Figure 5 demonstrate the results of using SSD to diagnose WLI and NBI esophageal neoplasm images, respectively. The SSD uses a marked bounding blue box to identify esophageal low-grade dysplasia, a gray box to identify high-grade dysplasia, and an orange box to identify esophageal cancer. The normal area does not display the frame, and the green box indicates the ground truth manually circled. Whether SSD can diagnose esophageal cancer is determined by comparing the ground truth and the bounding box.

### 4.3. Statistical Analysis

Fisher’s exact test was used to compare the diagnostic accuracy of SSD in using WLI and NBI for the detection of esophageal neoplasm, while a *p* value below 0.05 was considered statistically significant. To compare the diagnostic accuracy of SSD in using WLI and NBI in differentiating histological grade of esophageal neoplasm, the kappa coefficient was used to assess interrater diagnostic agreement between SSD and pathological diagnoses. All statistical operations were performed using STATA 15 statistical software.

## 5. Conclusions

Our SSD system showed good diagnostic performance in detecting neoplasm as well as classifying histological grade. An AI system might have potential in diagnosing the early stages of esophageal neoplasms without the necessity of magnified endoscopy and an experienced endoscopist. It is hoped that early and accurate diagnosis of esophageal neoplasm will provide a less harmful therapeutic option and improve patient prognosis.

## Figures and Tables

**Figure 1 cancers-13-00321-f001:**
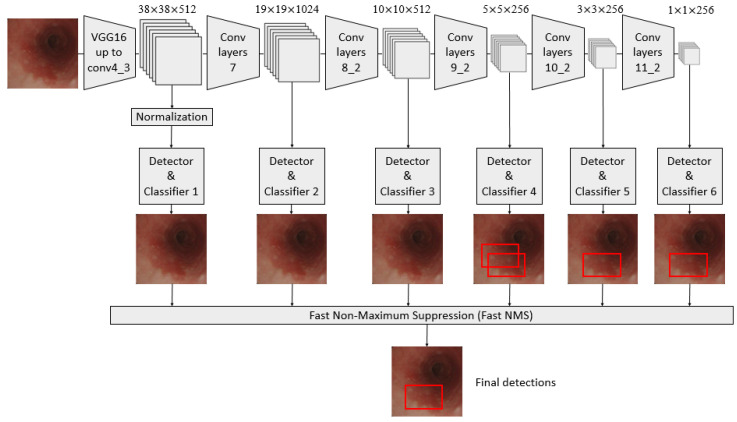
Schematic flowchart of the single-shot multibox detector (SSD) for esophageal neoplasms. Under the input of 300 × 300 SSD, Conv7, Conv8_2, Conv9_2, Conv10_2, and Conv11_2 were extracted as the feature maps for detection in the newly added convolutional layer to generate more layers with smaller scales and facilitate multilayer feature fusion. A total of six feature maps whose sizes were (38, 38), (19, 19), (10, 10), (5, 5), (3, 3), and (1, 1) were extracted with the Conv4_3 layer.

**Figure 2 cancers-13-00321-f002:**
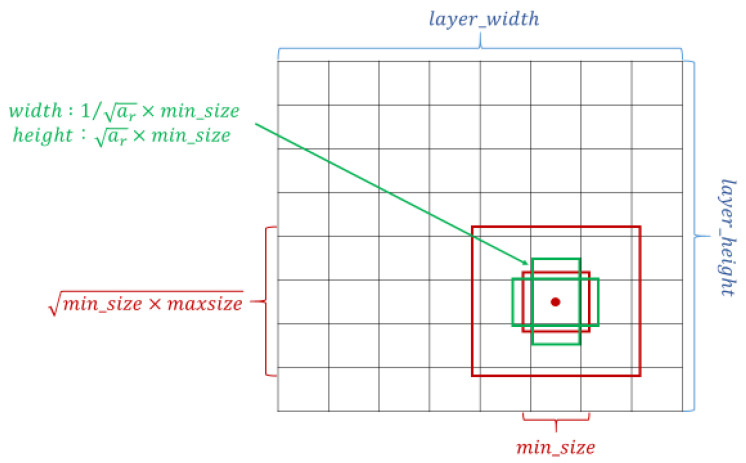
Schematic of the settings of the default box. Taking the midpoint of each point on the feature map as the center, a series of square prior frames with different sizes and the same center point was generated. With the square prior frame, the ground truth cannot be all square. If the square prior frame was used to predict the ground truth, the prediction effect could not be optimized. Therefore, in addition to the square prior frame, multiple rectangular prior frames needed to be added to match the ground truth and increase the model prediction effect.

**Figure 3 cancers-13-00321-f003:**
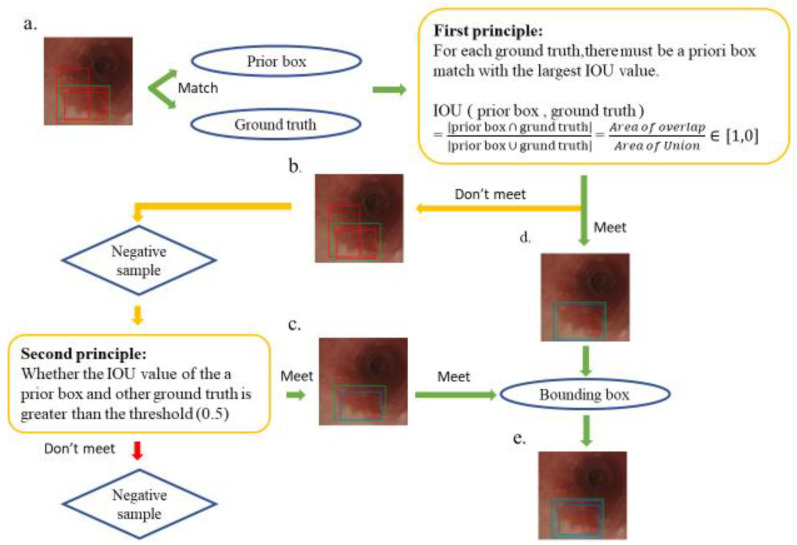
Schematic of the principle of matching ground truth with a prior box. The green box in the esophagus image is the ground truth, the multiple red boxes are prior boxes, and the blue box is the boundary box that finally matches the ground truth. (**a**) A match between multiple prior frames and ground truth; (**b**) a negative sample that does not meet the first principle during the matching process; (**d**) a positive sample that conforms to the first principle, which later becomes a boundary box; (**c**) a positive sample that does not meet the first principle but meets the second principle and also becomes a bounding box; and (**e**) the final output bounding box, which conforms to the first and second principles, which is the final SSD prediction.

**Figure 4 cancers-13-00321-f004:**
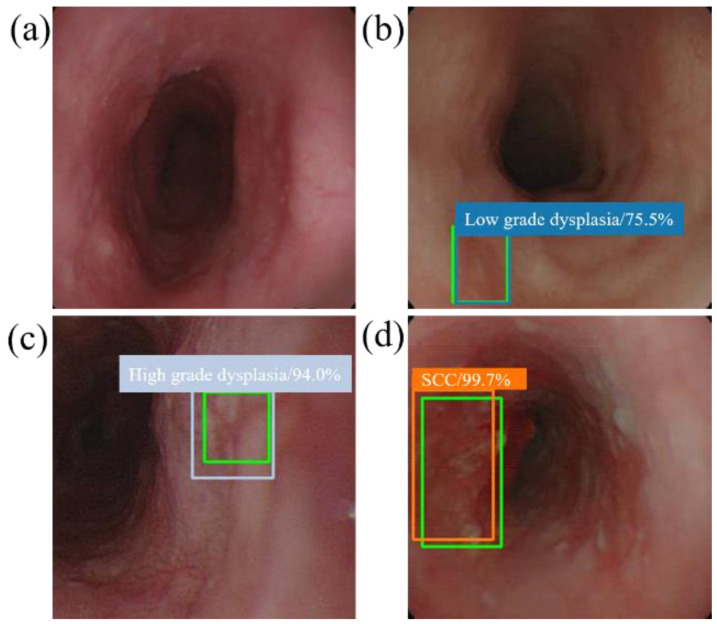
SSD diagnostic results of WLI esophageal neoplasm image. All green boxes in the figure are ground truth. The numbers in the label indicate the probability of being judged as the number of esophageal cancer stages in the box. (**a**) Image of a normal esophagus. No frame is displayed under the SSD diagnosis; thus, the SSD diagnosis was normal esophagus. (**b**) A blue border box is displayed around the lesion area, determined as low-grade dysplasia. (**c**) A gray border box is displayed around the lesion area, determined as high-grade dysplasia. (**d**) Esophageal endoscopy image with an esophageal cancer area. Under SSD diagnosis, an orange bounding box surrounds the lesion area and determines that the lesion area is cancer.

**Figure 5 cancers-13-00321-f005:**
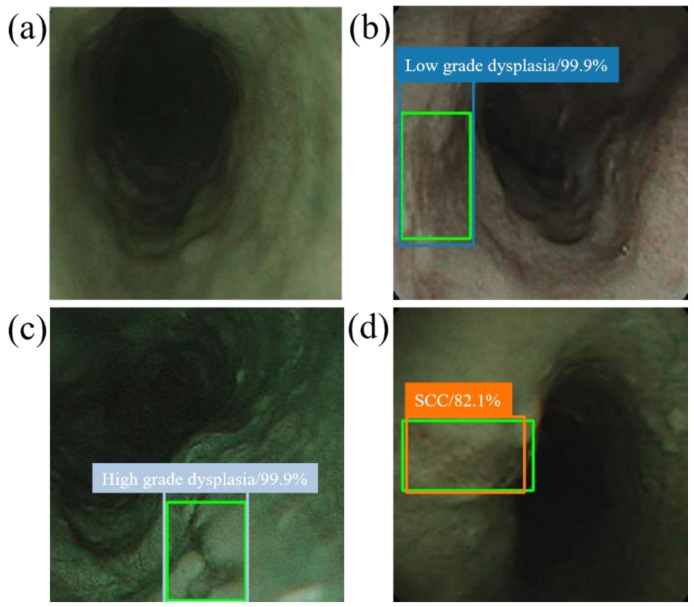
SSD diagnostic result of NBI esophageal neoplasm image. All green boxes in the figure are ground truth. The numbers in the label indicate the probability of being judged as the number of esophageal cancer stages in the box. (**a**) Image of a normal esophagus. No frame is displayed under the SSD diagnosis. Thus, the SSD diagnosis was normal esophagus. (**b**) Esophageal endoscopy image with a low-grade dysplastic area. Under SSD diagnosis, a blue border box is displayed around the lesion area, and the lesion area is therefore determined to be low-grade dysplasia. (**c**) Esophageal endoscopic image with an area of high-grade dysplasia. Under SSD diagnosis, a gray border box is displayed around the lesion area, and the lesion area is therefore determined to be high-grade dysplasia. (**d**) Esophageal endoscopic image with an esophageal cancer area. Under SSD diagnosis, an orange bounding box surrounds the lesion area and determines that the lesion area is cancer.

**Table 1 cancers-13-00321-t001:** Detailed results of 264 esophageal images diagnosed by our single-shot multibox detector (SSD).

Diagnostic Result	SSD Diagnosis	
	Normal	Neoplasm
Pathological Diagnosis		
Comprehensive		
Normal	38	16
Neoplasm	8	202
WLI		
Normal	13	4
Neoplasm	5	90
NBI		
Normal	25	12
Neoplasm	3	112

Abbreviations: SSD, single-shot multibox detector; WLI, white-light image; NBI, narrow-band image.

**Table 2 cancers-13-00321-t002:** Diagnostic performance of SSD for esophageal neoplasm.

Diagnostic Performance	WLI	NBI	Comprehensive
Accuracy (%)	92.0	90.1	90.9
Sensitivity (%)	94.7	97.4	96.2
Specificity (%)	76.5	67.6	70.4
PPV (%)	95.7	90.3	92.7
NPV (%)	72.2	89.3	82.6

Abbreviations: SSD, single-shot multibox detector; WLI, white-light image; NBI, narrow-band image; PPV, positive predictive value; NPV, negative predictive value.

**Table 3 cancers-13-00321-t003:** Detailed results of SSD in diagnosing different histological grades of esophageal neoplasms.

Diagnostic Result	SSD Diagnosis			
	Low-Grade Dysplasia	High-Grade Dysplasia	Cancer (SCC)	
**Pathological diagnosis**				Accuracy
Comprehensive				92%
Low-grade dysplasia	26	2	3	
High-grade dysplasia	2	68	8	
Cancer (SCC)	0	1	92	
WLI				89%
Low-grade dysplasia	11	1	3	
High-grade dysplasia	1	30	4	
Cancer (SCC)	0	1	39	
NBI				95%
Low-grade dysplasia	15	1	0	
High-grade dysplasia	1	38	4	
Cancer (SCC)	0	0	53	

Abbreviations: SSD, single-shot multibox detector; WLI, white-light image; NBI, narrow-band image; SCC, squamous cell carcinoma.

**Table 4 cancers-13-00321-t004:** Diagnostic performance of SSD for different histological grades of esophageal neoplasm.

Diagnostic Performance	Sensitivity (%)	PPV (%)	F1-Score (%)
Comprehensive			
Low-grade dysplasia	83.4	92.8	88.1
High-grade dysplasia	87.2	95.8	91.3
Cancer (SCC)	98.9	89.3	93.9
WLI			
Low-grade dysplasia	73.3	91.7	81.5
High-grade dysplasia	85.7	93.8	89.6
Cancer (SCC)	97.5	84.8	90.7
NBI			
Low-grade dysplasia	93.8	93.8	93.8
High-grade dysplasia	88.4	97.4	92.7
Cancer (SCC)	100.0	93.0	96.4

Abbreviations: SSD, single-shot multibox detector; WLI, white-light image; NBI, narrow-band image; PPV, positive predictive value; SCC, squamous cell carcinoma.

## Data Availability

Data sharing not applicable.

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
