# Peer review of "Endoscopic Images by a Single-Shot Multibox Detector for the Identification of Early Cancerous Lesions in the Esophagus: A Pilot Study"

_cancers, 2021, doi:10.3390/cancers13020321_

Round 1

Reviewer 1 Report

I believe that the authors have significantly made changes mentioned in the previous report. The draft is now in an acceptable form and ready for further processing.

Author Response

We appreciated the reviewer’s comment. We spent a lot of time on this article.

Reviewer 2 Report

Revised manuscript evaluated the diagnostic potential and usability of an AI-assisted image analysis system in differentiating different grades of esophageal neoplasm. Authors have improved manuscript content following its revision.

COMMENTS:

Since authors indicated, even in their cover letter response that it is a pilot study, I would suggest they also indicate that in the manuscript title by adding “: A pilot study” to the current title.

Secondly, because of the organization of the manuscript content and presence of figures within the “Material & Methods” section, I would suggest they move the “Material & Methods” section so that it immediately follows the Introduction section. This is because of their unconventional inclusion of a lot of figures in “Material & Methods” section; which is typically not the case. Else, authors may have to move Figures 4 & 5 to results section (the other figures 1-3 can still be accommodated in “Material & Methods” section as they are more indicative of the process methodology) and provide a subsection heading for WLI & NBI esophageal neoplasm SSD Images.

Either way, the conclusion section ought to immediately after the discussion section and not after Material & Methods.

Based on Table 4 information, the indicated values in line 104 are for sensitivity and not PPV. Otherwise, it appears authors have a mix-up of labelling between sensitivity and PPV in table 4. Recheck information.

Author Response

Revised manuscript evaluated the diagnostic potential and usability of an AI-assisted image analysis system in differentiating different grades of esophageal neoplasm. Authors have improved manuscript content following its revision.

COMMENTS:

  1. Since authors indicated, even in their cover letter response that it is a pilot study, I would suggest they also indicate that in the manuscript title by adding “: A pilot study” to the current title.

Reply:

Thank you for your suggestion. We had revised our title as “Endoscopic images by a single-shot multibox detector for the identification of early cancerous lesion in the esophagus: A pilot study”.

  1. Secondly, because of the organization of the manuscript content and presence of figures within the “Material & Methods” section, I would suggest they move the “Material & Methods” section so that it immediately follows the Introduction section. This is because of their unconventional inclusion of a lot of figures in “Material & Methods” section; which is typically not the case. Else, authors may have to move Figures 4 & 5 to results section (the other figures 1-3 can still be accommodated in “Material & Methods” section as they are more indicative of the process methodology) and provide a subsection heading for WLI & NBI esophageal neoplasm SSD Images.

Either way, the conclusion section ought to immediately after the discussion section and not after Material & Methods.

Reply:

Thank you for your suggestion. Because the published model of cancers is organized by introduction-results-discussions-materials and methods-conclusions, we do not know is it appropriate to change the sequence.

  1. Based on Table 4 information, the indicated values in line 104 are for sensitivity and not PPV. Otherwise, it appears authors have a mix-up of labelling between sensitivity and PPV in table 4. Recheck information.

Reply:

Thank you for your correction. We have corrected this mistake in revised manuscript.